# Both Direct and Indirect Evidence Contribute to Dative Alternation Preferences in Language Models

**Qing Yao,**[τ] **Kanishka Misra,**[τ] **Leonie Weissweiler,**[μ] **Kyle Mahowald**[τ]

Department of Linguistics, The University of Texas at Austin,[τ]
Department of Linguistics and Philology, Uppsala University[μ]

qyao@utexas.edu, kmisra@utexas.edu, leonie.weissweiler@lingfil.uu.se
kyle@utexas.edu

## Abstract

Language models (LMs) tend to show human-like preferences on a number of syntactic phenomena, but the extent to which these are attributable to direct exposure to the phenomena or more general properties of language is unclear. We explore this with the English dative alternation (DO: *gave Y the X* vs. PO: *gave the X to Y*), using a controlled rearing paradigm wherein we iteratively train small LMs on systematically manipulated input. We focus on two properties that affect the choice of alternant: length and animacy. Both properties are directly present in datives but also reflect more global tendencies for shorter elements to precede longer ones and animates to precede inanimates. First, by manipulating and ablating datives for these biases in the input, we show that direct evidence of length and animacy matters, but easy-first preferences persist even without such evidence. Then, using LMs trained on systematically perturbed datasets to manipulate global length effects (re-linearizing sentences globally while preserving dependency structure), we find that dative preferences can emerge from indirect evidence. We conclude that LMs' emergent syntactic preferences come from a mix of direct and indirect sources.

## 1 Introduction

Consider the dative alternation. Roughly the same real-world event is conveyed by *She gave the boy who signed up for class and was excited it* (using the Double Object or DO form) and *She gave it to the boy who signed up for class and was excited* (the Prepositional Object or PO form). Yet most people would prefer the latter. The reason is that users of English typically show a "short-first" preference in the dative alternation. That is, if the theme is longer than the recipient they prefer the DO. If the theme is shorter, the PO. See an illustration of this in Fig. 1. Besides length, there are a host of other factors that influence the choice of alternation used, such as animacy and pronominality (Bresnan et al., 2007; de Marneffe et al., 2012).

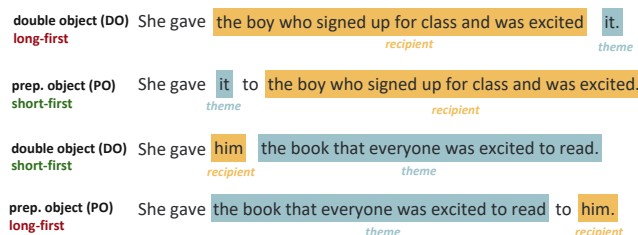

Figure 1: The dative alternation allows for either PO or DO realization. We highlight the arguments (theme and recipient) that are relevant to our feature analysis.

These preferences in the dative alternation are learned well by even relatively small LMs (Hawkins et al., 2020; Misra & Kim, 2024). But what has to be true of the training data for these preferences to emerge? Recent work has considered the role of indirect evidence in language learning, asking whether LMs learn syntactic phenomena by memorizing only the most similar instances or whether they rely on structurally similar instances in order to generalize. Evidence so far has been mixed, with some work arguing for generalization from indirect evidence. Misra & Mahowald (2024) find that a somewhat rare syntactic structure is learned, in part, from indirect evidence from more common related structures. Patil et al. (2024) find that grammatical generalizations are possible even when never encountered. Oba et al. (2024), though, find that indirect evidence is less important than direct evidence. Houghton et al. (2025) find that binomial preferences (e.g., "peanut butter and jelly" vs. "jelly and peanut butter") in models largely depend on memorization, with little evidence of generalization to human-like preferences for novel binomial pairs. Jumelet et al. (2024) find that language models can make some generalization about adjective order preference (e.g. "big black boxes" vs "black big boxes") on adjective pairs not seen in training.

We posit that the dative alternation is an elegant testing ground for exploring these questions, since it is linguistically well-studied and shown to be sensitive to a variety of factors that are not only directly observable in extant dative data, but which also depend on more general constraints in the language. One possibility (direct evidence) is that LMs prefer dative constructions that are highly similar to those observed in training. That is, if *She gave it to the boy who signed up for class and was excited* is attested in the training but *She gave the boy who signed up for class and was excited it* is not, then LMs might assign higher probability to strings that are very similar to the former. A second possibility (indirect evidence) is that the observed dative preferences are emergent from more general features of the input data. Specifically, we hypothesize that the LMs' preference for "short-first" and "animate-first" dative utterances might follow not from memorizing instances of such datives but from those preferences existing more generally in the data (Behaghel, 1932; MacDonald, 2013) – e.g., Tur et al. (2025) find contemporary LMs to follow the general "short-before-long" ordering across a variety of syntactic structures.

To adjudicate between these possibilities, we use a controlled rearing paradigm (Jumelet et al., 2021; Frank, 2023; Misra & Mahowald, 2024; Feng et al., 2024; Patil et al., 2024; Leong & Linzen, 2024; Kallini et al., 2024; Xu et al., 2025), whereby we train small models on carefully controlled input data and then test whether the predicted preferences emerge. We first train a `default` model on a roughly 88M-word corpus and test whether these preferences emerge in the dative. We then manipulate the input, balancing length and animacy-related statistics in the training set to remove any order bias in the instances of the dative construction. We do this by creating a matching PO for every DO, and a matching DO for every PO. We show that dative preferences still persist—although they get weaker. So we train another model just like `default` except all datives are inserted in the reverse alternant, such that the direct evidence is now the opposite of what it normally would be. We find that this neutralizes the standard dative preferences. Then, we train models which do not have direct exposure to datives, and see that length preferences still emerge.

The above results show that direct evidence matters but also that human-like trends emerge even without direct evidence, especially for length.[1] To ascertain whether these trends come from more global statistical patterns, we turn to manipulating global preferences directly. We do this by systematically reordering all constituents in training sentences based on their length, in order to create input corpora with systematically controlled length effects. We find that, while the usual dative length preferences emerge in the "short-first" corpus, the preferences are gradually lost as training sentences increasingly become more "long-first". We take this as evidence that **more general properties of the language non-trivially contribute to the observed dative preference in LMs**. As such, this work joins a larger body of work suggesting the emergence of abstract linguistic behavior in LMs from statistical regularities in their training (see Futrell & Mahowald, 2025, for a summary).

---

[1]Code and data for the experiments are available at https://github.com/dounick/dativelm.git

## 2 Data, Methods, and Models

### 2.1 Corpus

Our training data is a subset of the 2023 BabyLM corpus with 100M words (Warstadt et al., 2023), which we choose for its tractable and human-scale size. We exclude the QED portion, since it includes special symbols and lacks proper sentence breaks, making dative detection difficult. This leaves our initial training set at around 88M words.

### 2.2 Defining and Detecting Datives

Our approach crucially depends on identifying dative utterances in the corpus, so that we can manipulate them during training and also create a separate test set of dative utterances. Defining datives is not straightforward and has been handled in different ways in the literature (Levin, 1993; Bresnan & Nikitina, 2009; Liu & Morgan, 2020; Liu & Wulff, 2023). In particular, delineating datives requires making choices like whether to include benefactives (*baked me a cake*) or non-alternating verbs (*\*dedicated the mayor the park*).

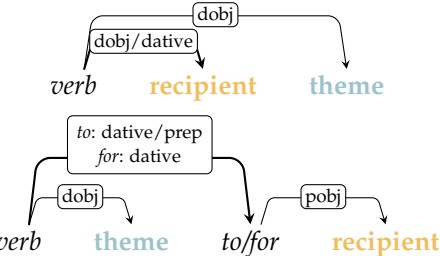

Figure 2: spaCy parses of possible DO datives (top) and PO datives (bottom).

First, we obtained dependencies parses from spaCy (Honnibal et al., 2020) for all utterances in the BabyLM dataset, on which the following definition of datives in the main paper is based. Then, we label an utterance as a DO dative if it contains a verb whose dependency contains a direct and an indirect object. For PO datives, we consider utterances that have a verb with a dependency to a direct object (the theme), and dependencies to an indirect object (the recipient) of the following form: 1) a *to* prepositional phrase under a `dative` or `prep` dependency; 2) a *for* prepositional phrase under a `dative` dependency. See Fig. 2 for an illustration.

This definition is intentionally broad, with intent to cover all constructions of interest, but may include false positives, such as the PO-like *burning it to the ground*. Precision is less important than recall in controlled rearing experiments, since recall is key for controlling dative exposure in our models. More details, including an alternative (stricter) method for detecting datives and a demonstration that our results do not depend heavily on the precise operationalization of datives, can be found in App. F.

**Recall Error Estimate and Artificial Pollution** To confirm that no datives slipped through, we spot-check a random sample of 4000 utterances classified as non-dative of token length greater than 5 by hand. We find just one dative entry, namely *they gave the poor professor divers and sundry medicines*, due to spaCy interpreting *professor divers* as a compound noun rather than an archaic spelling of *diverse*. From this, we estimate the recall rate to be 0.025% (compare with 2.57% of all entries in BabyLM being detected as datives under our loose but high-recall criteria). Considering that our training sets consists of around 9M non-dative utterances, we estimate there are 2250 false negative datives, as a conservative upper bound.

Following Misra & Mahowald (2024), we artificially "pollute" our data with counterbalanced data intended to offset the effect of the undetected datives. These undetected datives could, in principle, provide direct evidence for human-like dative preferences. To counteract them, we insert datives, in roughly equal number, that violate the standard preferences. Specifically, for each model where we manipulate just the dative exposure, we insert 2250 counterfactual datives (swapping the structure of attested datives) to minimize the effects of the estimated 2250 false negative datives. We estimate that the undetected datives are twice as likely to be a DO than a PO, so 1500 counterfactuals are inserted in the DO form (when attested was PO) and 750 in PO form (when attested was DO). For each model, we will discuss if counterfactual insertion is done as they are introduced. See Table 7 in the Appendix for a breakdown of the number of controlled dative exposures, estimates of unaccounted-for datives, and insertions of counterfactuals by model.

| Exp. | Model | Pre-manipulation | Post-manipulation |
|---|---|---|---|
| All | `default` | N/A | N/A |
| 1a | `balanced` | I gave the dog a bone | I gave the dog a bone
I gave a bone to the dog |
| 1b | `swapped-datives` | I gave the dog a bone | ~~I gave the dog a bone~~
I gave a bone to the dog |
| 1c | `no-datives` | I gave the dog a bone
He eats the green melon from the shop with a fork | ~~I gave the dog a bone~~
He eats the green melon from the shop with a fork |
| | `no-2postverbal` | I gave the dog a bone
He eats the green melon from the shop with a fork | ~~I gave the dog a bone~~
~~He eats the green melon from the shop with a fork~~ |
| 2 | `short-first` | He uses a fork to eat the green melon from the shop | [he] uses [a fork] [[to] eat [[the] [green] melon [from the shop]]] |
| | `random-first` | He uses a fork to eat the green melon from the shop | [[to] eat [[the] [green] melon [from the shop]]] uses [a fork] [he] |
| | `long-first` | He uses a fork to eat the green melon from the shop | [[[from the shop] [the] melon [green]] eat [to]] uses [a fork] [he] |
| | `long-first-headfinal` | He uses a fork to eat the green melon from the shop | [[[the shop **from**] [the] [green] **melon**] [to] **eat**] [a **fork**] [he] **uses** |

Table 1: An overview of the manipulations for our experiments. The `default` model is the same across all experiments. All models in Experiment 2 have no entries containing verbs with two postverbal arguments. Heads are in bold for the `long-first-headfinal` model.

## 2.3 Model Architecture

We use the OPT-125M architecture (Zhang et al., 2022). See App. A for specific training details. Each of our LMs is trained using the `transformers` library (Wolf et al., 2020). We train each LM with three random seeds to account for randomness in weight initialization. For models with exposure to datives, we ensure that the total number of datives (including estimated unaccounted-for datives and counterfactual insertions) is equal across models, so that we can fairly compare dative preferences across models.

## 2.4 Measuring Dative Preference

In all our subsequent analyses, we will specifically study our trained LMs' preferences between the two possible realization of a dative construction given a dative verb and its arguments, following previous works (Hawkins et al., 2020; Misra & Kim, 2024). Given a pair of DO and PO realizations of the same dative construction, we compute the preference of the LM as the difference between its log probabilities for the DO and the PO sentence, each normalized by its sentence length, to account for the extra *to/for* in the PO sentences. All length-normalized log-probabilities are computed using `minicons` (Misra, 2022).

## 2.5 A Test Set for Testing Preferences by Feature

To quantify our LMs' dative preferences across sentences with varying length and animacy features, we construct a test set consisting of pairs of dative-alternating sentences sampled from the BabyLM test corpus (again excluding the QED portion). We sample 1000 sentences detected as a DO dative and 1000 detected as PO from the test corpus. The recipients and themes are labeled for pronominality using `spaCy`, and manually labeled for animacy by us. Both features are considered to be binary in our analysis.

We manually ensure that each instance is a dative, and that the verb usage is considered to alternate *in general*, even though the constructed unattested alternant may not sound natural. This is because we are interested in isolating the differences in the models' preferences according to differences in the features of interest (e.g., length and animacy), so the inclusion of potentially unnatural utterances is precisely what we are interested in.

To create DO forms from attested PO forms, we remove the preposition and swap the position of the recipient and theme. Creating PO sentences from attested DO datives is more complicated since we consider both *to*-datives and benefactives (i.e., with *for*) to be

part of our PO dative set. While some dative verbs have fixed preposition usage, some can use either preposition depending on indirect object (recipient/beneficiary) and theme: *bring that letter to me* vs. *bring a napkin for me*. Taking this into account, we take the following steps to create our PO sentences: first, we check if the verb occurs as a dative/benefactive verb in Levin (1993). If it does, then we check if it alternates, and based on whether its alternation is listed as a *to*-dative or a benefactive, we decide the preposition (*to* vs. *for*). For the remaining verbs not classified as datives in Levin (1993) or not listed as alternating, we use the Llama-3.2 3B model (Grattafiori et al., 2024) to decide between the appropriate surface form by creating sentences in both forms, and then selecting the one which has the higher log-probability according to the model. Using this method gives us 2000 pairs of datives in our test set, with a total of 76 distinct verb lemmata.

**Feature Encoding**  For future regression analyses, we compute a length difference score and an animacy difference score for each entry in the test set. Length difference is defined as the log difference in length between the recipient and theme. Because animacy is binary-coded, their difference scores take on values of $-1$, $0$, or $1$. An animacy difference of $0$ means that either the theme and recipient are both animate or both inanimate, a difference of $1$ means that the recipient is animate and theme is inanimate, and a difference of $-1$ means that the recipient is inanimate and the theme is animate.

The noun phrases in the attested datives are approximately in the natural distribution of features in DO and PO datives since they are randomly sampled. The average log difference in length between the recipient and theme is $-0.886$ in DOs and $0.199$ in POs, consistent with the easy-first preference. Animates are strongly aligned with the recipient position in either alternant (recipients were animate in 95.1% of DOs, and 71% of POs), consistent with the fact that inanimates are bad recipients (Beavers, 2011).

## 3  Preliminary Experiment: Does a BabyLM-trained LM learn human-like dative preferences?

While many dative verbs alternate freely between the DO and the PO, human alternation behavior for these verbs is graded (Bresnan & Nikitina, 2009). As a preliminary experiment, we test the extent to which our BabyLM-trained LMs capture human-like dative preferences in general. Specifically, we train an LM on our subset of the BabyLM corpus (see §2.1), which we will refer to as **default**. The training set is filtered to consist of 66,822 datives in both DO and PO forms (and possibly undetected datives). The model should have roughly equal exposure to each alternant.

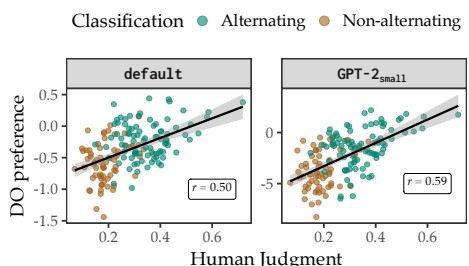

Figure 3: Pearson's correlation between human judgments and predictions from **default** and **GPT-2**small LMs, both z-scored, across the 155/200 verbs in *DAIS* (Hawkins et al., 2020) that are present in the BabyLM corpus. **GPT-2**small predictions are taken directly from *DAIS*.

For testing, we use the *DAIS* dataset from Hawkins et al. (2020), which contains 5000 different pairs of sentences in the DO and PO forms, spanning 200 different verbs. In addition, it also contains behavioral results from LMs such as **GPT-2**small and other models. We only select the sentences for which the dative verb appears in a dative construction in the training set, giving us 155 unique verbs and 3672 pairs of sentences. For each instance, the dataset also has a measure of the human judgments of preference for DO over PO sentences, analogous to our main measure of DO preference (see §2.4). To compare our model's preferences to those from humans, we followed Hawkins et al. and compared the average Human DO preference for the 155 target verbs to the average (z-scored) LM DO preference by computing their Pearson correlation.

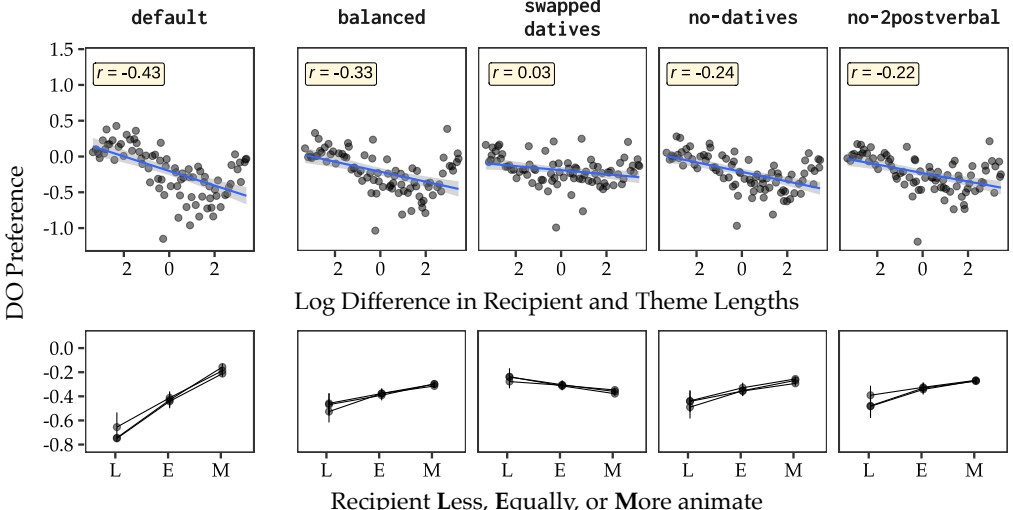

Figure 4: Correlation between DO preference and length difference (**top**) and animacy difference (**bottom**) by model on instances in the test set ($N = 2000$). A negative correlation indicates short-first preference, while greater DO preference for recipients being animate indicates animate-first preference. The seeds with best evaluation loss are shown for length, whereas all three seeds are shown for animacy. Models shown here are for Experiment 1.

**Clear Preferences Emerge**   We compute the correlations with human judgments obtained from our model and, for reference, we also re-compute the correlation obtained for GPT-2small (Radford et al., 2019) for the subset of the verbs we included in our analysis. As seen in Fig. 3, our model (with seed selected by best evaluation loss) obtains a correlation of $r = 0.50$ with the human ratings on *DAIS*. Not only is this positive, it is not far off from the correlation obtained by GPT-2small ($r = 0.59$), even though our model was trained on a corpus that is several orders of magnitude smaller than that of GPT-2small.

To measure the effect of length on dative preference, we compute the log-difference in length between the theme and recipient (positive value means recipient is shorter), and examine its correlation with the DO preference. As expected, we observe a negative correlation between the length difference and the DO preference: $r = -0.43$ for the default model with the best loss—i.e, the model prefers DO more when the recipient is shorter than the theme. We also found an animacy-first trend, such that DO preference was highest when the recipient was more animate than the theme.

We fit a mixed-effect model predicting DO preference based on fixed effects of length difference and animacy difference. We include random intercepts for verb lemma and seed with maximal random effect slope structures, excluding correlations to help convergence, following Barr et al. (2013). We compute p-values using nested model comparisons. For the default model, we found significant effects of length ($\hat{\beta} = -.174$, $p < .0001$) but not for animacy ($\hat{\beta} = .065$, $p = .11$). On further inspection, we found that the observed animate-first effects seem to be driven by specific verbs—e.g., *leave it to me* over *leave me it*, but *write me a letter* over *write a letter to me*. We include a discussion about verb-specific effects in App. E. We show length and animacy effects for default (and subsequent models) in Fig. 4.

## 4   Experiment 1: Balancing, Swapping, and Removing Datives

Having established that our small models show clear dative preferences and, in particular, show a strong preference for length, a key question emerges: are these preferences a result of memorization from similar dative sentences, or do they emerge from a more general property of the input? To address this, we run a series of controlled rearing experiments, in which we systematically modify the evidence that the model receives and examine the persistence of the length and animacy effects.

In Experiment 1a, we manipulate the dative sentences in training to be perfectly balanced, effectively **removing all length and animacy effects from dative constructions exposed to the model.** In Experiment 1b, we use the same training set as for `default`, except all datives are in swapped-order (i.e. recipient and theme are "out-of-order"). **This means that the model will receive the opposite direct evidence that we expect in the `default` case.** In Experiment 1c, we then entirely remove dative constructions, as well as all constructions with two postverbal arguments, so that the models do not receive any direct evidence. We also artificially pollute (see §2.2) the training set to approximately neutralize the effect of undetected datives in Experiments 1a and 1c (skipping it in Experiment 1b since the vast majority of datives in its training set are in reversed order anyway).

Broadly, we assess the significance of our manipulations in two ways. The first is that we fit a mixed effect regression separately for each model, predicting DO preference based on length and animacy (as described above). This tells us whether, in each model, we observe length and animacy effects significantly different from 0 (Table 2 for all experiments). The second broad analysis is a global mixed effect model predicting DO preference across all experiments, based on fixed effects of length and animacy and the interact of each with model, treating `default` as baseline, as described in App. C. The interaction of each term (length and animacy) with model tells us whether our manipulations result in length/animacy effects significantly different from the `default` model (Table 4).

## 4.1 Experiment 1a: Effect of Balancing Dative Alternating Forms

The goal of our first manipulation is to balance the datives by matching the PO and DO frequencies and then creating a PO utterance to match every DO utterance. A model trained on such manipulated data should not observe any direct associations between word order and the features of the dative verb or recipient and theme. In particular, this means that the dataset will have no preference for the short argument to come first, and no preference for animates to come first (since we balance every "short-first" sentence with the counterfactual "long-first" sentence, and similarly for any animacy bias).

We balance the dataset by creating alternate forms for each occurrence of a dative, using the method outlined in §2.5. For instance, for the DO sentence *I gave the dog a bone* we create its PO form *I gave a bone to the dog* and then add both sentences in the training set (see Table 1). We ensure that the LMs trained on this manipulated corpus still have roughly the same number of datives in total as `default` after artificial pollution by balancing 32,850 attested DOs and 32,850 attested POs with their unattested alternants. By design, this model should observe no word-order preferences in datives. We refer to this model as `balanced`. As can be seen in Fig. 4, the length effect is weakened in the `balanced` results but still very much present: the correlation goes from $r = -0.43$ in the `default` to $r = -0.33$.

| Exp. | Model | Length | Animacy |
|------|-------|--------|---------|
| All | `default` | $-0.174$*** | $0.065$ |
| 1a | `balanced` | $-0.081$*** | $0.032$ |
| 1b | `swapped-datives` | $-0.012$ | $-0.036$ |
| 1c | `no-datives` | $-0.048$* | $-0.007$ |
| | `no-2postverbal` | $-0.056$** | $-0.035$ |
| 2 | `short-first` | $-0.064$** | $0.004$ |
| | `random-first` | $-0.017$ | $0.007$ |
| | `long-first` | $0.010$ | $0.068$ |
| | `long-first-headfinal` | $0.055$*** | $0.004$ |

Table 2: Regression coefficients from mixed effect models, fit separately for each LM's DO preference, with * for $p < .05$, ** for $p < 0.01$, and *** for $p < .001$. Length and animacy are difference scores, as described in §2.5.

As shown in Table 4, our global regression model showed that the effect of balancing significantly weakened the length and animacy effects relative to `default`. But, as shown in Table 2, using the same per-model regression as before but focusing on the `balanced` model's data, we again found a significant effect of length (but not for animacy). We also once again found verb-specific animacy effects.

## 4.2 Experiment 1b: Effect of Reversing All Datives

We see that when a model receives preference-neutral dative input in training, easy-first association still arises, albeit to a lesser extent. Taking this a step further, we train a `swapped-datives` model where all datives shown to the model are out-of-order from their

natural order. Its training set is identical to that of `default`, except the controlled 66,822 DO and PO datives are inserted in their unattested alternant forms. Extrapolating from the previous models' preferences, we should expect `swapped-datives` to have even less short-first preferences, if not reversed.

This is indeed the case: `swapped-datives` has a roughly neutral length effect of $r = 0.03$. Animacy effects are reversed, suggesting swapping direct evidence has an effect on dative preferences. `swapped-datives` showed significantly smaller length and animacy effects than the `default` LMs (see Table 4). Length and animacy coefficients were not significant by our per-condition regression analysis (see Table 2). Thus the `swapped-datives` models, as expected, finds effects in the opposite direction of `default`, but the magnitude is smaller.

### 4.3 Experiment 1c: Effect of Removing Dative and Ditransitive-like Constructions

We have shown that dative preferences persist even when datives are balanced in training, and are only roughly neutralized when all dative exposure are counterfactuals. This suggests that non-dative sentences are contributing to the learned preferences. If the models had no exposure to datives whatsoever, can they still recover dative preferences? This case is crucially different from the balanced case: in the balanced case, the models get direct evidence that there is no length or animacy effect in datives. In the case where all datives are removed, there is no evidence either way, so any preferences must come from elsewhere. To test this, we train an LM on a version of our BabyLM corpus without datives, which we refer to as `no-datives`.

Additionally, it is possible that our LM extracts dative preferences from preferences within constructions that share the structure with the dative. For example, other verb alternations such as *spray/load* (Levin, 1993, Section 2.3): *Jack sprayed the paint on the wall* vs. *Jack sprayed the wall with the paint* and creation/transformation (Levin, 1993, Section 2.4): *Martha carved a toy out of the piece of wood* vs. *Martha carved the piece of wood into a toy* could all be subject to similar easy-first biases, and the model can plausibly identify them with datives. For this reason, we also train an LM which has no exposure to any ditransitives or cases where the verbs take a prepositional object (not restricted to *to/for* prepositions) as the second object—which we refer to as `no-2postverbal`, i.e., cases with two postverbal arguments. To perform this ablation, we remove all entries which meet our dative criterion but in addition consider all prepositions instead of only considering *to/for*.

As with balanced datives, the length and animacy effects are reduced compared to the `default` model. As shown in Fig. 4 and Table 2, we again find a persistent length effect: $r = -0.24$ for `no-datives` and $r = -0.22$ for `no-2postverbal`. While there is not significant evidence for a global animacy-first preference, we again observe verb-specific effects that are interestingly similar to the effects in the `default` case (see Fig. 8 in App. E).

### 4.4 Interim Discussion

Overall, two different methods of changing the input data (balancing and removing datives) fail to remove the length effect, although in all cases, the effect is unsurprisingly weaker than in the `default` model. Only when all the input datives are in reversed order do length effects become closer to neutral (though interestingly they are not reversed). The effects of animacy are more nuanced. Here, we do not find a global "animate-first" preference—instead, this preference seems to depend on the verb (see App. E). For instance, when the recipient is more animate, *leave* and *sell* are strongly preferred in the PO, while *bring* and *write* show strong DO preferences. At the same time, the verb-specific effects are similar between the `default` and `no-datives` condition (Fig. 8), suggesting some higher order effect of indirect evidence. We leave deeper exploration of this to future work.

## 5 Experiment 2: Effect of Global Length Manipulation

Having shown that length effects, in particular, persist even when we remove datives or balance them, a natural question emerges: Where do these effects come from? One

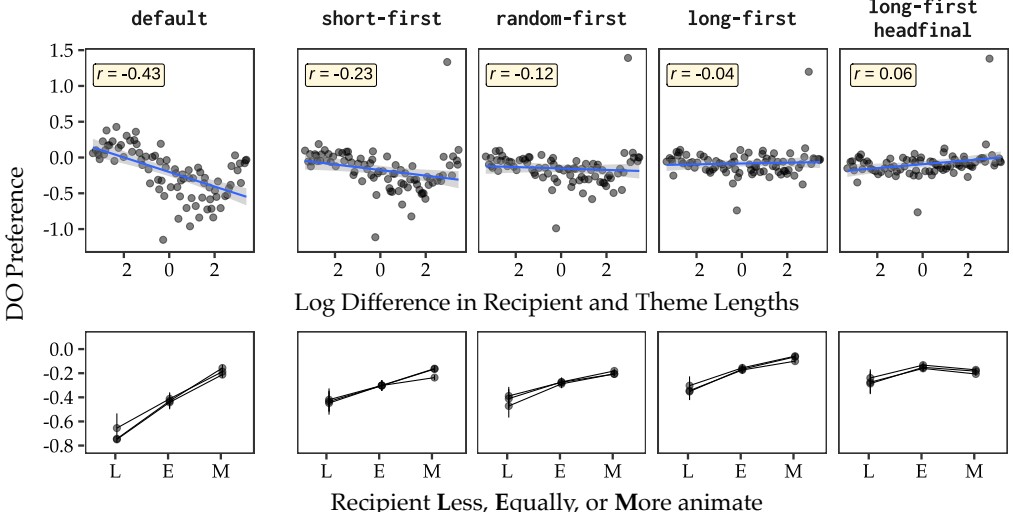

Figure 5: Correlation between DO preference and length difference (**top**) and animacy difference (**bottom**) by model on instances in the test set ($N = 2000$). Models shown here are for Experiment 2.

hypothesis is they come from a more global linguistic preference for short constituents to appear earlier in English. In this experiment, we then ask whether more global properties of the input affect length preferences by systematically manipulating the training corpus to either favor placing short syntactic constituents first or long ones first, or—as a control— place either long or short-first randomly.

To perform our global length manipulations, we operate on our `no-2postverbal` corpus that excludes datives and related constructions. For each sentence, the children of any node in its dependency representation are ordered by length (either short-first or long-first). Since a long-first preference has been observed in head-final languages (Yamashita & Chang, 2001; Futrell et al., 2020), we also create a corpus where constituents are arranged by decreasing length and every head appears in the final position. See App. H for pseudocode. We train a `short-first`, `long-first`, and `long-first-headfinal` LM on these corpora, respectively. As a control, we also train a model on a corpus where child-constituent balancing is randomly chosen (`random-first`). The training sets for these four models are identical apart from constituent ordering, and so no counterfactual pollution is used.

The training set of `no-2postverbal` is already strongly short-first – roughly 70% of entries are short-first in their original form and 44% long-first. Disregarding simple sentences where no head has more than one child (since these are automatically short/long-first), 53% of sentences are already short-first compared to 16% for long-first. This suggests that preferences learned by the `short-first` model should be similar to that of `no-2postverbal`. We provide a more detailed analysis of how short-first the original corpus is in App. I.

The length-manipulated LMs have no exposure to any cases with two postverbal arguments, and in addition to that, they observe a biased version of English word order with consistent length properties. Therefore, if global length properties have an effect, we should expect a gradient such that the dative length effect is greatest in the `short-first` manipulation, minimal in the `random-first` manipulation, and non-existent or reversed in the `long-first-headfinal` manipulation.

## 5.1 Results

As shown in Fig. 5, `short-first`, `random-first`, `long-first`, and `long-first-headfinal` show a length-effect gradient, appearing in the expected order. To assess significance of these results, we ran another regression (similar to our global regression, see App. C for details and Table 5 for results), focused on just the length-manipulated conditions and with the `random-first` condition as baseline. As expected, the `long-first` and

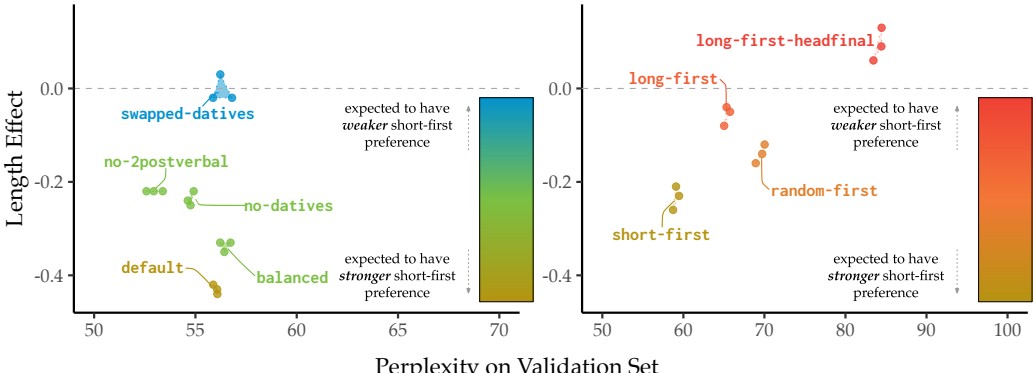

Figure 6: Correlation of DO preference score with log-difference in length (y-axis) vs. the geometric mean of perplexities (x-axis) on a fixed sample of 10,000 sentences in the validation set (with appropriate length manipulations if applicable). **Left:** Models without length manipulations; **Right:** Models with length manipulations. The more **golden** a model's color, the more we expect its training set manipulation to favor short-first.

**long-first-headfinal** models showed significantly weakened length preferences compared to **random-first**, but **short-first** showed significantly stronger length preferences than **random-first**. Therefore, when no direct evidence is available, the learned dative preferences tend towards preferences in the hypothesized indirect evidence.

Since the training data in many of our experiments are heavily manipulated, it is possible that observed length effects are due to higher perplexities from less good models overall. That is, maybe some models just don't learn dative preferences very well because they don't learn *anything* very well. To test this, we compute the geometric mean of perplexities on 10,000 sentences from the validation set with corresponding length manipulations for every model. These perplexity values and length effect coefficients per seed are shown in Fig. 6. The observed perplexities do not fully explain the patterns in our data. For example, the **long-first** models have a lower perplexity on the validation set than **random-first** models, and yet they show weaker (less negative) length effects. We see that within groups of models with or without length manipulation, the more **blue** or **red** models have a weaker short-first preference regardless of perplexities. This suggests that length effects are due to fundamental changes in global constituent ordering, beyond those captured by perplexity.

## 6    Conclusion

Bespoke small models learn reasonably human-like preferences for the dative alternation. Length preference is persistent even when we train models on datasets without direct evidence for such a preference, suggesting that a short-first preference is a more general property of English. By then training models on counterfactual languages where the indirect evidence becomes decreasingly "short-first", we show that the length effect is gradually lost, indicative of the role of indirect evidence in acquiring that preference. The story for animacy is more subtle. Instead of a global animate-first effect, we found verb-specific differences where other factors such as the theme might also play a role. At the same time, these verb-specific effects were persistent even in the absence of direct evidence, and oftentimes did not flip in ablations that flipped the direct evidence.

Overall, we take these results to show that LMs' dative preferences emerge from more general properties of the language, not just those directly observable in dative constructions. As such, we join a growing consensus that a key property of the success of LMs is the consistency of linguistic evidence in the input.

## Acknowledgments

We thank John Beavers, Sasha Boguraev, Richard Futrell, and the UT Computational Linguistics research community for discussions and feedback. Qing Yao thanks Simon Todd for

his mentorship during his undergraduate thesis, which extended into this work. Qing Yao is supported by the Donald D. Harrington Fellowship at UT Austin. Kanishka Misra is supported by the Donald D. Harrington Faculty Fellowship at UT Austin. Leonie Weissweiler was supported by a postdoctoral fellowship of the German Academic Exchange Service (DAAD). Kyle Mahowald was supported by NSF CAREER grant 2339729 from the Director of STEM Education (EDU) Division of Research on Learning in Formal and Informal (DRL).

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

## A   Training Details

Our models are trained largely following the procedure in Misra & Mahowald (2024). We have a total of eleven model-types, of which nine are discussed in the main text (with two following a more strict definition of datives discussed in App. D), and for each type, we conducted three training runs with different seeds to account for variation due to initialization. In total, this amounts to 33 LM training runs. See Table 3 for a summary of training details—hyperparameters, compute hardware, etc.

## B   List of Models on Huggingface

Huggingface paths to three seeded runs of each model and their corresponding training sets are shown below, where **x**∈{strict_default, loose_default, strict_balanced, loose_balanced, swapped-datives, no-datives, no-2postverbal, short-first, random-first, long-first, long-first-headfinal}.

- Model path: `qing-yao/{x}_seed-{21,42,63}_1e-3`
- Dataset path: `datasets/qing-yao/datives-{x}`

| Hyperparameter | Value |
|---|---:|
| Architecture | OPT (Zhang et al., 2022) |
| Embed size | 768 |
| FFN dimension | 3072 |
| Num. layers | 12 |
| Attention heads | 12 |
| Vocab size | 32768 |
| Max seq. length | 256 |
| Warmup steps | 32000 |
| Total parameters | 110M |
| Compute | 1x NVIDIA A40 |
| Training time | 11 hours |

Table 3: Training details

## C   Regression Models

### C.1   Per-model Regressions

For each LM, we treat length difference and animacy difference as predictors for predicting DO preference scores across three seeds, and run the following regression model to obtain $\hat{\beta}$ coefficients for length and animacy.

The logic of this regression, which we run separately for each model, is to ask whether length and animacy effects are significantly different from 0.

```
l_full <- score ~ length_difference + animacy_difference +
(1 + length_difference + animacy_difference | verb_lemma) +
(1 + length_difference + animacy_difference | seed)
```

Here, length difference and animacy difference are fixed effects. Verb lemma has random intercepts and slopes for both fixed effects, with correlations, and seed has random intercepts only.

We compute significance scores for length difference and animacy difference in each LM is via a nested model comparison, using the R command: `anova(l_full, l_reduced)`, where `l_reduced` excludes the fixed effects related to the predictor.

## C.2 Global Regression Models

We also fit a global model, in which we treat **default** as a baseline and print its output below. In this model, we used random intercepts for verb lemma and random seed, with random slopes for length difference and animacy difference. We excluded other random slopes and correlations for convergence.

The logic of this regression is to ask whether length and animacy effects, for each model, differ significantly from the length and animacy effects in the default model. We use the `lmerTest` package to compute significance (Satterthwaite's method).

In Table 4, the interaction coefficients (e.g., **balanced** : Length Difference) can be thought of as telling us how much the Length or Animacy difference differs in the given condition, as opposed to the **default** baseline.

```
l_global <- score ~ condition*length_difference +
condition*animacy_difference +
(1 + length_difference + animacy_difference || verb_lemma) +
(1 + length_difference + animacy_difference || seed)
```

| Predictor | Estimate |
|---|---|
| Intercept | $-0.459$*** |
| **balanced** | $0.046$*** |
| **no-datives** | $0.078$*** |
| **no-2postverbal** | $0.087$*** |
| **swapped-datives** | $0.121$*** |
| **short-first** | $0.127$*** |
| **random-first** | $0.139$*** |
| **long-first** | $0.249$*** |
| **long-first-headfinal** | $0.248$*** |
| Length Difference | $-0.189$*** |
| Animacy Difference | $0.091$*** |
| **balanced** : Length Difference | $0.105$*** |
| **no-datives** : Length Difference | $0.135$*** |
| **no-2postverbal** : Length Difference | $0.140$*** |
| **swapped-datives** : Length Difference | $0.227$*** |
| **short-first** : Length Difference | $0.131$*** |
| **random-first** : Length Difference | $0.187$*** |
| **long-first** : Length Difference | $0.225$*** |
| **long-first-headfinal** : Length Difference | $0.266$*** |
| **balanced** : Animacy Difference | $-0.117$*** |
| **no-datives** : Animacy Difference | $-0.104$*** |
| **no-2postverbal** : Animacy Difference | $-0.109$*** |
| **swapped-datives** : Animacy Difference | $-0.195$*** |
| **short-first** : Animacy Difference | $-0.070$*** |
| **random-first** : Animacy Difference | $-0.066$*** |
| **long-first** : Animacy Difference | $-0.034$* |
| **long-first-headfinal** : Animacy Difference | $-0.126$ |

Table 4: Summary of estimates from model comparing to **default** with significance levels indicated by asterisks (***$p < 0.001$, **$p < 0.01$, *$p < 0.05$).

For Experiment 2, we fit another regression (same as the global regression above) to just the length-manipulated data in Experiment 2, with **random-first** as the baseline. The logic of this regression is to test whether the length-manipulated models are significantly different from the **random-first** models in length effects. (We also compute animacy effects here, but have no predictions about them for this set of models.)

| Predictor | Estimate |
|---|---|
| Intercept | − 0.307*** |
| `short-first` | − 0.012 |
| `long-first` | 0.110*** |
| `long-first-headfinal` | 0.109*** |
| Length Difference | − 0.005 |
| Animacy Difference | 0.002 |
| `short-first` : Length difference | − 0.055*** |
| `long-first` : Length Difference | 0.039*** |
| `long-first-headfinal` : Length Difference | 0.080*** |
| `short-first` : Animacy Difference | − 0.004 |
| `long-first` : Animacy Difference | 0.032* |
| `long-first-headfinal` : Animacy Difference | − 0.060*** |

Table 5: Regression output focusing on just the length manipulations, with the `random-first` condition as a baseline (***$p < 0.001$, **$p < 0.01$, *$p < 0.05$).

## D  Strict-dative `default` and `balanced` Results

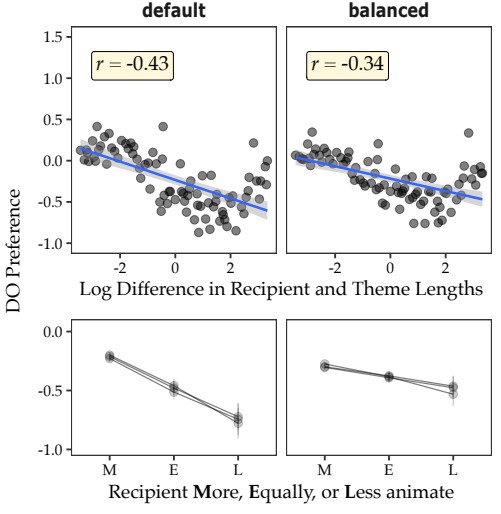

| Exp. | Model | Length | Animacy |
|---|---|---|---|
| All | `default` | − 0.174*** | 0.062 |
| 1a | `balanced` | − 0.072*** | 0.029 |

Table 6: Regression coefficients from mixed effect model, fit separately for strict `default` and `balanced` predicting DO preference, with *** for $p < .001$.

Figure 7: Correlation between DO preference and length difference and animacy difference for strict `default` and `balanced`.

We refer to the datives defined in the main paper as loose-datives, and introduce a stricter notion of datives, which we term strict-datives. These are proper subsets of loose-datives. Here we take a stricter approach and only consider loose-datives whose verbs satisfy the following conditions, based on the classification laid out by Levin (1993): 1) it is classified as a *to*-dative or a benefactive verb; and 2) its usage in the detected entry is consistent with its alternation behavior (e.g. verb is classified as PO only but is detected in a DO, verb is classified to be benefactive but is used in a *to*-dative).

This definition results in 125,415 DOs and 66,822 POs, compared to 139,249 DOs and 118,040 POs under the loose definition.

We now show that the choice between strict and loose classifications has little effects on our analysis by repeating Experiment 1a under the strict notion.

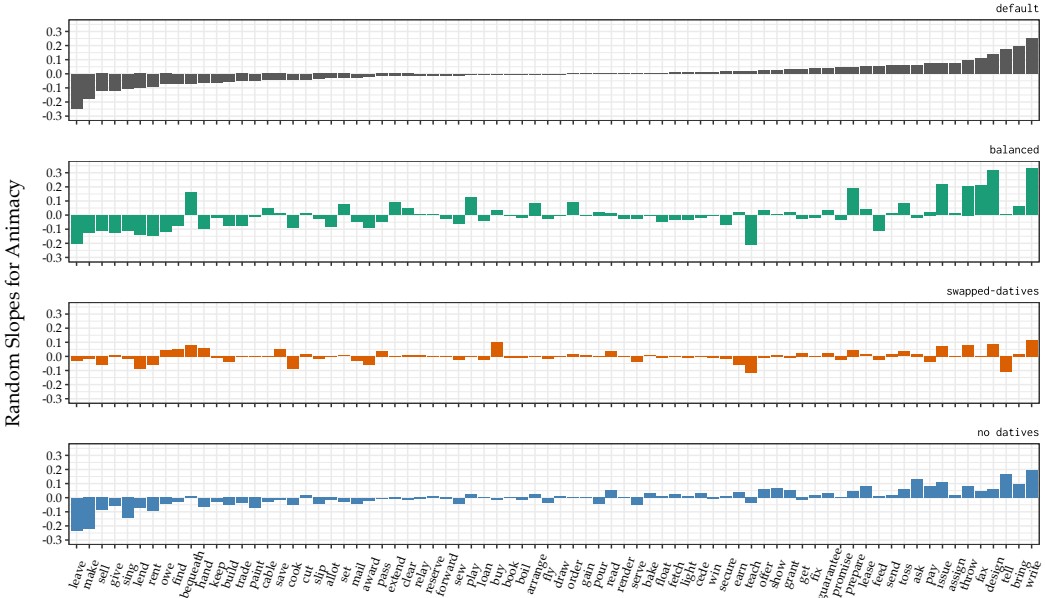

Figure 8: Verb-specific random slope-differences (differences from the global random slope) for the animacy term in our linear mixed-effects model analysis. Positive value indicates relatively greater DO preference for animate recipients, while negative ones indicate lower DO preference for inanimate recipients.

Both the regression coefficients in Table 6 and the length and animacy effects in Fig. 7 for the strict models are similar to their loose counterparts. We therefore focus on just the loose notion in the main paper, so that the ablated models are more aggressively removing direct evidence.

## E  Verb-specific Random Slopes for Animacy

Random slopes by verb lemma for **default**, **balanced**, **swapped-datives**, and **no-datives** are shown in Fig. 8. We see that animacy effects for certain verbs are heavily affected by the manipulations in **balanced** and **swapped-datives**, indicating that the animacy preferences of these verbs are more tied to direct evidence. However, **no-datives** recovers similar, albeit a bit weaker, animacy effects as in **default**. This suggests that usages of dative verbs in the indirect evidence can inform an LM's dative animacy preferences.

## F  Detecting Datives and 2postverbals

Using spaCy, we first detect all utterances that could contain a DO or a PO dative. For DOs, we search for utterances containing a verb which has a dative and a dobj dependency or two dobj dependencies. For POs, we look for utterances containing a verb with a dobj dependency, and with either a *to* prepositional phrase under a dative or prep dependency or a *for* prepositional phrase under a dative dependency. From these entries, we perform a sanity check to ensure that the noun phrases come after the verb, and the recipient and theme have appropriate parts of speech.

These entries are then loose-datives. From these entries, we check if the verb lemma is classified as a dative or benefactive verb in Levin (1993). If so, we check if it is in the permissible form (e.g. if the verb is classified as benefactive only, the entry cannot be a *to*-PO, or if the verb is classified as DO only, the entry cannot be a PO). If a loose-dative satisfies the above, it is also a strict-dative.

The entries which are not detected as loose-datives are not necessarily non-datives, since no small set of definitions based on dependency parses can capture all datives. To ensure high recall, the non-dative set is obtained by removing ambiguous entries from the set of non-loose-datives. Via a manual check, we found that undetected datives generally include verbs having both a dobj and a clausal complement dependency (as in *So you're telling me this is over?*) and entries with verbs having a dobj and a prep arc to *for* with a pobj dependency (as in *He played 38 games for Japan until 1971*). The remaining non-loose-dative entries after this procedure are considered non-datives.

Lastly, we obtain the set of non-2postverbal from the set of non-datives by further removing entries containing a verb with two direct objects of any kind, and entries containing a verb with a dobj and a prep-pobj of any kind.

An implementation of this procedure is in the code repository under `src/detect_datives.py`

## G  Artificial Pollution

The counts of dative exposure by model are listed in Table 7.

| Type | | default | balanced | no-datives | no-2postverbal |
|---|---|---|---|---|---|
| Controlled datives | DO | 66822 | 65700 | 0 | 0 |
| | PO | 66822 | 65700 | 0 | 0 |
| False negatives (estimate) | DO | 1500 | 1500 | 1500 | 1500 |
| | PO | 750 | 750 | 750 | 750 |
| Counterfactuals | DO | 0 | 1500 | 1500 | 1500 |
| | PO | 0 | 750 | 750 | 750 |
| Total (estimate) | DO | 68322 | 68700 | 3000 | 3000 |
| | PO | 67572 | 67200 | 1500 | 1500 |
| % of total utterances | | 1.55 | 1.55 | 0.05 | 0.05 |

Table 7: Number of dative exposures by model. False negatives are estimated based on an error rate of $1/4000$. Length-manipulated models are the same as **no-2postverbal** apart from constituent ordering, and **swapped-datives** is the same as **default** apart from controlled datives being in their unnatural alternants.

## H  Manipulating Constituent Ordering by Length

We use the following algorithm to reorder all constituents of a sentence to short/long-first using spaCy dependency parses. Sentences for the **random-first** model are created by randomly choosing short/long-first when sorting the children of each node.

---

**Algorithm 1** Length Manipulation in Syntax Tree

---

1: **function** BUILDNODE(token, visited, short_first, head_final)
2:     **if** token ∈ visited **then**
3:         **return** visited[token]
4:     **end if**
5:     Recursively process child nodes
6:     **if** short_first **then**
7:         Sort children by constituent length ascending
8:     **else**
9:         Sort children by constituent length descending
10:     **end if**
11:     Reassign original positions to maintain relative ordering
12:     **if** head_final **then**
13:         Set current node's constituent as the concatenation of its children constituents, with the node itself at the end
14:     **else**
15:         Set current node's constituent as the concatenation of its children constituents
16:     **end if**
17:     Store node in visited dictionary
18:     **return** node
19: **end function**

---

These algorithms can be found in the code repository under `src/utils.py`.

## I   Measuring the Short-Firstness of English

We provide a more fine-grained measurement of the degree to which English sentences are short-first or long-first in terms of the number of adjacent swap-operations required to rearrange the sentence in length-sorted order.

For each sentence, we compute the inversion number corresponding to a head that has $n \geq 2$ dependents as the number of out-of-order pairs (w.r.t. to short-first or long-first). The inversion number of a head is precisely equal to the number of adjacent swaps to sort its dependents. As such, a head with $n$ dependents has an inversion number at most $\binom{n}{2}$.

We then add up the inversion numbers of each head within the sentence, and normalize by the maximum possible sum of inversion numbers (attained when dependents of every head are maximally out-of-order) to obtain a measurement of short/long-firstness.

For entries where constituent rearrangements were possible (i.e. there exists one head with two or more dependents), it takes 23.8% of the maximal number of swaps to reach short-first, compared to 53.2% for long-first. This means that, on average, it takes more than twice the number of adjacent-swap operations to sort by long-first compared to short-first.

