# OpenReview forum: "Both Direct and Indirect Evidence Contribute to Dative Alternation Preferences in Language Models"
_colmweb.org/COLM/2025/Conference — COLM 2025_

### Official Review · Reviewer_BPUy · 2025-04-23

**Rating:** 8
**Confidence:** 3
**Ethics Flag:** 1

**Summary:**

This draft investigates the process of learning English dative construction by means of language models. The draft looks into the alternation between double object and prepositional object in relation to two features: length and animacy of the arguments. Many experiments are presented, and the main finding is that preferences emerge from a mix of direct exposure to the phenomena and more general properties of the language.

**Questions To Authors:**

l.126: Each of the our LMs

**Reasons To Accept:**

Experiments have been carefully designed and discussed.
Interesting and novel results on an already established line of research.

**Reasons To Reject:**

I do not see any reason to reject.

---

> ### Author Response · Authors · 2025-05-29
>
> Thank you for this positive assessment! We will fix the noted typo.

---

### Official Review · Reviewer_M82d · 2025-04-25

**Rating:** 8
**Confidence:** 4
**Ethics Flag:** 1

**Summary:**

This paper explores the behavior of LMs on the English dative alternation. First, the author ensures that a "babyLM" shows roughly the same behavior as larger LMs on this alternation. They then go on to do controlled rearing experiments, to manipulate the evidence, i.e., the datives seen during training, to explore the preference of the model, and the impact of length and animacy preferences, as well as indirect evidence. The paper is very well written and clear, with well-motivated and interesting experiments, shedding some light on the dative alternation, as well as on the impact of general language preferences.

**Reasons To Accept:**

The paper explores interesting questions of language preferences in a sound setup, leading to interesting conclusions. I especially find it interesting that you manage to investigate both indirect and direct evidence for the different preferences.

The controlled rearing method is interesting, since it can provide causal evidence. The use in this paper is sound and interesting, and could potentially provide inspiration for future studies of other phenomena.

**Reasons To Reject:**

No major reason to reject

A couple of minor issues (which should not affect acceptance, but which I did not find a good place to put):

* In the conclusion, you draw general conclusions about language, even though you just experimented on English. Please rephrase (it is clear that everything is for English in the rest of the paper, though).
* Please try to make the figures and tables larger in the final paper. They are now hard to read.
* Since you manually verified 2000 samples in 2.5, it would have been preferable also to report how accurate this sample was

---

> ### Author Response · Authors · 2025-05-29
>
> Thank you for this positive assessment! In response, we will be clearer in the conclusion about the English focus, and make the figures and tables larger.
>
> Regarding the 2000 samples: we iteratively sampled until we had 2000 but did not keep track of the overall sample required to get to that level. As a ballpark, we would estimate that roughly 50% of the initial entries were “good” and so we filtered ~4000 (very rough estimate) down to 2000.The left-out entries were not all false positives, but also included entries which are non-alternating.

---

> ### Comment · Area_Chair_Gg7z · 2025-06-03
> **Could you engage with or just acknowledge the authors' response?**
>
> Thank you!

---

> ### Comment · Reviewer_M82d · 2025-06-04
> **Reviewer answer**
>
> Thank you for the clarification about the 2000 samples.
>
> I stand by my initial review and think this is a good paper.

---

### Official Review · Reviewer_BYEk · 2025-05-11

**Rating:** 8
**Confidence:** 4
**Ethics Flag:** 1

**Summary:**

This paper uses the controlled rearing methodology to assess the degree to which LMs rely on direct and indirect evidence when choosing to output double object or prepositional object forms of dative sentences. That is, they train LMs on various different datasets, manipulating the extent and types of direct / indirect evidence available at training time; they then evaluate LM preferences. They find that even when all direct evidence concerning when one or another form of dative should be output has been removed from the training corpus, LMs still exhibit a preference for shorter constituents first (animacy effects are weaker). They then re-train on data where length ordering is altered, and find that constituent length ordering evidence does influence LMs' preferences re: dative alternation. They conclude that models rely on both direct and indirect evidence.

**Questions To Authors:**

- 1b and 1c are swapped in Table 1
- 186: Section 2.1, did you mean?
- Figure 3 is a bit tough on readers, as it is referenced a couple of times later in the text as well. Maybe split it up, but have full version in the appendix for comparison?
- 342-343: This would be clearer if written in the active voice

**Reasons To Accept:**

In general, I think this is a strong paper, for the following reasons:
- The controlled rearing methodology is one of the best ways of getting causal evidence that a certain phenomenon in the training data led to a given behavior on the part of the LM. It takes quite a bit of effort to do this, both in terms of curating the training data and training the actual models, and the authors have trained multiple seeds for each LM.
- This paper seems well grounded in the human psycholinguistics literature, although I'm not an expert on this particular phenomenon.
- This paper is in general quite well-polished and readable

**Reasons To Reject:**

- Experiment 2 is, in my view, less convincing than experiment 1. This is because the corpora in experiment 2 are unnatural / composed of lots of data that is not grammatically correct English. As a result, one wonders if the effects observed are really only a result of this rather dramatic shift in the corpus, as opposed to the more subtle length ordering rules that exist in English in reality. I think the ideal experiment here would be to reorder only those constituents that are allowed to be reordered. That is, in the long-first case, "He uses a fork to eat the green melon from the shop" could become "To eat the green melon from the shop, he uses a fork", but not "from the shop the melon green eat to uses a fork he". I would expect effect sizes to shrink somewhat. That said, I think this sort of complex reordering would be quite difficult to do, so I understand why the current experiment was done instead.
- The framing / motivation for this project is a bit weak.
    - The papers cited in favor of the "direct evidence only" view are not that convincing from a methodological point of view, and also not very influential. I mention the latter point because I think there are more influential voices arguing that LLMs' abilities come in large part from copying / direct evidence. I realize that these are murky waters to wade into, but engaging with higher-level debates around LLMs could help emphasize what is at stake here.
    - Similarly, I expected to see some sort of discussion of these results in relation to humans, and was a bit disappointed to find that missing.
    - Relatedly, there have been a few papers now using controlled rearing to show how LLMs use direct and indirect evidence to learn various constructions. I agree that this is now a "growing consensus", but I'm also left wondering what the bigger picture is. I'm now left wondering what the bigger picture is - why does this matter, and what next?

---

> ### Author Response · Authors · 2025-05-29
>
> Thank you for engaging so deeply with our paper! We are glad you found our use of the controlled rearing paradigm to be compelling, and are grateful for your detailed suggestions–to which we respond below:
>
> 1. On experiment 2’s unnaturalness: We also wondered if the results could emerge from the fact that the corpora are ordered in a way that fundamentally departs from the general structure of the target language. This is exactly why we ran our perplexity vs. effects analysis, shown in Fig 4. If the overall un-naturalness is all that explained our results, then we’d have expected to see a strong correlation between perplexity and length effects. However, in Fig 4 we see that there are cases where perplexity was low but there was no length effect (swapped-datives) and also cases where perplexity was relatively higher (random first) but the length effects were non-trivially greater than those with lower perplexity (long-first). Therefore, insofar as perplexity captures general unnaturalness in the corpus, we don’t really see a strong effect. That said, we like your idea of focusing on only re-orderable constituents—something that would be of value in dependency manipulation studies more broadly. We will explore running such a study for the Camera Ready.
>
> 2. On engaging with broader debates about memorization and generalization: We agree it would be useful for the camera ready to strengthen the broader implications of this work. If you have a particular suggestion for “influential voices arguing that LLMs' abilities come in large part from copying / direct evidence”, please do let us know. Re the line of work by Carlini et al. on memorization: we do not necessarily dispute the claim that LMs can memorize verbatim. Our claim is instead that while they do memorize, there are interesting finer-grained generalizations that they can show, something that can emerge even without direct evidence in their training data. So while we agree that direct evidence is critical to LMs, it is not the full story. We agree that we can bring out these positions more and will do so in the Camera Ready.
>
> 3. On the bigger picture and relation to human processing and learning: we see controlled rearing as a powerful and intuitive method that allows us to understand fine-grained generalization behavior in statistical learners like LMs. We are, in general, optimistic about the portability of these kinds of results to humans. But we think future work will require explicitly merging these results with psycholinguistics. In particular, there have been cases where methods similar to controlled rearing have been applied to generate hypotheses about human learners by using more stringent controls, though this field is currently in its infancy. We would be happy to make this broader level connection in the future version of this paper, and we thank you for pushing us in this direction.

---

> > ### Comment · Reviewer_BYEk · 2025-05-30
> >
> > Thanks for these responses! Don't think I will raise my score, but I am satisfied by the rebuttal and remain excited about this paper.
> > 1) Agreed that the perplexity bit is a reasonable control, but my critique was a bit different from the "maybe some models just don’t learn dative preferences very well because they don’t learn anything very well" point raised in the paper. I'm more concerned about the fact that, while some of the datasets used for training are essentially subsets of the English language, many of the datasets used in this section are not grammatically valid English sentences at all. The more the training data diverges from natural English data, the less comfortable I am making claims about how LMs process natural language English
> > 2) Didn't have anything specifically in mind here, and I agree that this paper doesn't run contrary to e.g. Carlini et al.; rather, I thought it provided more evidence of when / how  / that LMs do sometimes generalize in linguistically interesting and predictable ways.
> > 3) Great, that sort of discussion would be helpful

---

### Official Review · Reviewer_tLzp · 2025-05-13

**Rating:** 6
**Confidence:** 4
**Ethics Flag:** 1

**Summary:**

The paper sets out to investigate whether the preferences observed in language models on a number of syntactic phenomena are directly attributable to exposure to the phenomenon or to general properties of language.

They apply this investigation to the dative alternation (I gave a book to Bill/I gave Bill a book), concentrating on length (it is ok to say 'I gave him a book' but much less good to say 'I gave the boy that you met yesterday and did not like a book', preferred option would be 'I gave a book to the boy that you met yesterday and did not like' and animacy  effects (animate recipients is much preferred as first argument).


The results are that a mixture of construction specific (for example, verb-specific) effects and general effects are at play.

Both the (rather vague, in my opinion) question and the linguistic phenomenon are much studied, as are the properties here investigated (order and animacy). The results are rather unsurprising, and confirm a lot of previous evidence on dependency length minimisation and animacy effects.

**Questions To Authors:**

Perhaps I can suggest exploring the more verb-dependent biases and try to develop methods to study semantic roles? Length and animacy (as expressed by pronouns, a property of English) have been widely studied. Studying deeper abstract properties might use the interesting method to greater insight.

**Reasons To Accept:**

- The controlled rearing  methodology is interesting and supports very well controlled experimentation
- The experimental set up is  very careful and controlled

**Reasons To Reject:**

I am afraid I found the paper rather thought-un-provoking, testing rather intuitive hypotheses.
I am afraid I think that the same results could have been gleaned by carefully analyzing attested dependency length effects and other work on datives

I don't feel I learnt much, despite the interesting method.

---

> ### Author Response · Authors · 2025-05-29
>
> Thanks for this review. We are glad you found the methods interesting and the experiments careful and controlled, even if finding the paper “rather thought-un-provoking”. We agree there is more to do here around the verb-dependent biases and think it’s well-spotted that this is fertile ground. In fact, we have an interesting series of mixed effect analyses of the verb biases, which we touch on in the paper but hope to expand on in the Camera Ready.
>
> While we agree such analyses will be interesting, we would also push back against the idea that they should be the core contribution of this paper since the controlled rearing paradigm is new enough that there is much more work to be done at this higher level of granularity, to evaluate the paradigm and understand its strengths and limitations. We see this work as setting the stage for future linguistic exploration of the kind you describe, using a well-studied linguistic phenomenon.
>
> Re whether results could be gleaned from analyzing dependency lengths: We also agree these results could be *predicted* by “carefully analyzing attested dependency length effects and other work on datives”, but the core contribution is to see if we get these (predicted) results using the controlled rearing paradigm. The predictability of the results is the point, in a way. While careful analysis of dependency effects allows us to answer questions about what cues exist or what behavioral effects are present in data, the controlled rearing paradigm allows us to answer this question purely from a generalization angle. That is, given the presence of (direct/indirect) cues, is a statistical learner able to pick up on them? In that sense we think these methods are complementary.
>
> So, while we see the proposed follow-ups as interesting, we don’t think it is a demerit of this paper that it focuses on the core phenomenon. And we would more generally advocate for not seeing it as a demerit of work, in general, if the results are “rather intuitive”. In fact, for a relatively new method in a fast-moving area, we see it as a positive!

---

> > ### Comment · Reviewer_tLzp · 2025-06-04
> >
> > Thank you for your reply. You say  that the predictability of the results is kind of the point of the paper, given that your goal is to demonstrate a novel method. I agree that if you want to make a methodological point, reproducing predictable results is the right way to go. Maybe the paper should make this methodological point clearer.
> >
> > I have raised my score.

---

> ### Comment · Area_Chair_Gg7z · 2025-06-03
> **Could you engage with or just acknowledge the authors' response?**
>
> Thank you!

---

### Decision · Program_Chairs · 2025-07-08

**Decision:**

Accept

**Comment:**

Dative sentences in English usually sound better if the first argument is shorter than the second (regardless of whether the first argument is the goal or the theme). This paper shows that small language models show this preference as well when trained on a natural corpus, and then explores ablations of the corpus to investigate the source of this preference. It finds that this preference emerges even when datives are excluded from training, suggesting that the short-first preference is language-wide.

This is a clear and methodologically well-executed paper, with clever controls (e.g., balancing the effect of possible failures to exclude particular constructions by "polluting" the corpus with noise in the opposite direction). I share the concerns of reviewer tLzp about the scientific conclusions, which seems fairly modest. It's not clear to me if anyone (from Behagel onwards) has argued that the short-before-long principle is specific to datives, so the experiments aren't contributing to a live debate. In the discussion, the authors respond to these concerns by framing the paper as showcasing a new methodology, but that doesn't seem convincing to me as the methodology is quite similar to Mishra and Mahowald (2024) and a cluster of similar papers (cited in this manuscript). I also thought reviewer BYEk's concerns with Experiment 2 were valid. As a minor point, I wonder if it is fair to say that models show human-like preferences when their correlation with human judgments is only r = 0.5.

Overall, although this paper is somewhat incremental, it is generally well executed and I don't see a risk in accepting it.